# Anti-LAMP-2 Antibody Seropositivity in Children with Primary Systemic Vasculitis Affecting Medium- and Large-Sized Vessels

**DOI:** 10.3390/ijms25073771

**Published:** 2024-03-28

**Authors:** Tayfun Hilmi Akbaba, Kirandeep K. Toor, Simranpreet K. Mann, Kristen M. Gibson, Gabriel Alejandro Alfaro, Banu Balci-Peynircioglu, David A. Cabral, Kimberly A. Morishita, Kelly L. Brown

**Affiliations:** 1BC Children’s Hospital Research Institute, Vancouver, BC V5Z 4H4, Canada; 2Division of Rheumatology, Department of Pediatrics, University of British Columbia, Vancouver, BC V6T 1Z4, Canada; 3Department of Medical Biology, Faculty of Medicine, Hacettepe University, 06800 Ankara, Turkey; 4Women+ and Children’s Health Sciences, University of British Columbia, Vancouver, BC V6T 1Z4, Canada; 5Department of Microbiology and Immunology, University of British Columbia, Vancouver, BC V6T 1Z4, Canada; 6Department of Medical Genetics, University of British Columbia, Vancouver, BC V6T 1Z4, Canada; 7Meso Scale Diagnostics, LLC, Rockville, MD 20850, USA; 8BC Children’s Hospital, Vancouver, BC V6H 3V4, Canada

**Keywords:** lysosome-associated membrane protein-2, anti-neutrophil cytoplasmic antibodies, childhood-onset primary vasculitis, autoantibodies, Takayasu’s arteritis, polyarteritis nodosa

## Abstract

Chronic primary systemic vasculitis (PSV) comprises a group of heterogeneous diseases that are broadly classified by affected blood vessel size, clinical traits and the presence (or absence) of anti-neutrophil cytoplasmic antibodies (ANCA) against proteinase 3 (PR3) and myeloperoxidase (MPO). In small vessel vasculitis (SVV), ANCA are not present in all patients, and they are rarely detected in patients with vasculitis involving medium (MVV) and large (LVV) blood vessels. Some studies have demonstrated that lysosome-associated membrane protein-2 (LAMP-2/CD107b) is a target of ANCA in SVV, but its presence and prognostic value in childhood MVV and LVV is not known. This study utilized retrospective sera and clinical data obtained from 90 children and adolescents with chronic PSV affecting small (SVV, n = 53), medium (MVV, n = 16), and large (LVV, n = 21) blood vessels. LAMP-2-ANCA were measured in time-of-diagnosis sera using a custom electrochemiluminescence assay. The threshold for seropositivity was established in a comparator cohort of patients with systemic autoinflammatory disease. The proportion of LAMP-2-ANCA-seropositive individuals and sera concentrations of LAMP-2-ANCA were assessed for associations with overall and organ-specific disease activity at diagnosis and one-year follow up. This study demonstrated a greater time-of-diagnosis prevalence and sera concentration of LAMP-2-ANCA in MVV (52.9% seropositive) and LVV (76.2%) compared to SVV (45.3%). Further, LAMP-2-ANCA-seropositive individuals had significantly lower overall, but not organ-specific, disease activity at diagnosis. This did not, however, result in a greater reduction in disease activity or the likelihood of achieving inactive disease one-year after diagnosis. The results of this study demonstrate particularly high prevalence and concentration of LAMP-2-ANCA in chronic PSV that affects large blood vessels and is seronegative for traditional ANCA. Our findings invite reconsideration of roles for autoantigens other than MPO and PR3 in pediatric vasculitis, particularly in medium- and large-sized blood vessels.

## 1. Introduction

Chronic primary systemic vasculitis (PSV) is an umbrella term for a family of heterogeneous diseases that are commonly characterized by inflammation and damage in blood vessel walls. PSV in children and adolescents has an average age of onset of 10–14 years and is particularly rare (<23/100,000 cases annually in North America) compared to the disease in adults (onset > 50 years) [1,2,3]. In both pediatric- and adult-onset vasculitis, different disease subtypes are broadly classified under the predominant size—small, medium, and large—of the affected blood vessels and consideration of differing clinical features, histologic analysis of affected tissues, and the presence (or absence) of anti-neutrophil cytoplasmic antibodies (ANCA) [4,5].

ANCA are a family of autoantibodies that can target distinct autoantigens in the cytoplasmic (c-ANCA) and perinuclear (p-ANCA) region of neutrophils and, to a lesser extent, monocytes [6]. ANCA seropositivity is predominantly observed in chronic PSV that affects small vessels, and in these cases, it targets one of two antigens: proteinase 3 (PR3) and myeloperoxidase (MPO). Seropositivity and specificity of ANCA for PR3 or MPO has some proven utility in the diagnosis and differentiation of “ANCA-associated vasculitis (AAV)” subtypes and more recently were demonstrated to have power in predicting disease-associated risks in adult-onset disease [4]. Similar diagnostic and prognostic tools do not exist for “ANCA-negative” vasculitis [7], which in children includes ~10–30% of cases of small-vessel vasculitis and the majority, if not all, forms of vasculitis affecting medium to large blood vessels [8].

Beyond PR3 and MPO, some studies have demonstrated that lysosome-associated membrane protein-2 (LAMP-2/CD107b) is an antigenic target of ANCA. LAMP-2 is a heavily glycosylated lysosome and plasma membrane protein that, in contrast to MPO and PR3, is expressed in almost every cell and tissue type in the body [9]. One epitope of LAMP-2 has 100% amino acid homology with type I fimbriated bacterial adhesion protein, FimH. Kain et al. [10] demonstrated that FimH-immunized rats produce ANCA against human LAMP-2 and spontaneously develop microvascular injury, glomerulonephritis, and lung damage. Previous reports have demonstrated an increase in LAMP-2 protein in sera from adults with a medium-sized vessel vasculitis subtype called polyarteritis nodosa (PAN) [11]. Elevated circulating concentrations of anti-LAMP-2 autoantibodies (LAMP-2-ANCA) have been observed in adults with small-vessel ANCA-associated vasculitis (AAV) and pauci-immune crescentic glomerulonephritis [12], as well as children with AAV [13]. Subsequent to this, elevated circulating concentrations of LAMP-2 protein [11,14] and LAMP-2-ANCA [13,15] were detected in adults with chronic PSV affecting medium- and large-sized blood vessels. Stemming from these collective reports, we hypothesized that LAMP-2-ANCA would be present in childhood-onset chronic PSV subtypes that affect medium to large blood vessels and have no detectable concentrations of circulating PR3-ANCA or MPO-ANCA.

## 2. Results

### 2.1. Greater Prevalence and Elevated Sera Concentration of LAMP-2-ANCA in PR3- and MPO- ANCA-Negative Vasculitis Subtypes Affecting Medium to Large Blood Vessels

A total of 90 children and adolescents newly diagnosed with chronic primary systemic vasculitis (PSV) affecting small- (small-vessel vasculitis, SVV, n = 53), medium- (MVV, n = 16), and large- (LVV, n = 21) sized blood vessels were included in this study (Table 1). The median age at symptom onset (13.3 years, range 1.9–17.8 years) and biological sex at birth (66.7% female) were comparable between individuals with vasculitis affecting vessels of different size: SVV median onset 14.2 years, 66.0% female; MVV median onset 11.2 years, 62.5% female; and LVV median onset 12.6 years, 71.4% female. Compared to individuals with a systemic inflammatory disease lacking autoimmune features (SAID cohort; see Methods), a greater percentage of participants with SVV, MVV, and LVV were LAMP-2-ANCA-seropositive, with the greatest prevalence observed in chronic PSV subtypes, where larger vessels were involved: 45.3% (24/53) of SVV patients are LAMP-2-ANCA-seropositive compared to 52.9% (9/16) in MVV and 76.2% (16/21) in LVV (Table 2). The concentration of sera LAMP-2-ANCA followed a similar trend in that a significantly higher concentration of LAMP-2-ANCA were detected in individuals with SVV (*p* = 0.0225), MVV (*p* = 0.0056) and LVV (*p* = 0.0001) compared to the SAID cohort (Figure 1A). Among participants with PSV, significantly higher concentrations of LAMP-2-ANCA were detected in individuals with MVV (2027.8 ng/mL, range 237.7–6210.9 ng/mL) compared to those with SVV (1197.1 ng/mL, range 134.0–4281.9 ng/mL) (*p* = 0.0207). Likewise, significantly higher concentrations of sera LAMP-2-ANCA were present in individuals with LVV (3965.7 ng/mL, range 103.8–12,413.7 ng/mL) compared to both SVV (*p* < 0.0001) and MVV (*p* = 0.0418). Focusing on seropositive participants only, the mean concentrations of LAMP-2-ANCA detected in our cohort were 1976.3 ng/mL in SVV, 3172.9 ng/mL in MVV and 5040.5 ng/mL in LVV.

Consistent with a greater number of LAMP-2-ANCA-seropositive individuals with MVV and LVV compared to SVV, LAMP-2-ANCA seropositivity was observed in the majority of individuals (23/35; 65.7%) seronegative for MPO-ANCA and PR3-ANCA, and by comparison, in fewer individuals seropositive for PR3-ANCA (11/30; 36.7%) (Table 2) or MPO-ANCA (9/18; 50%). Similarly, the concentration of LAMP-2-ANCA in individuals seronegative for MPO- and PR3-ANCA was significantly higher compared to LAMP-2-ANCA titers in PR3-ANCA- (*p* = 0.0008, Figure 1B) and MPO-ANCA-positive patients (*p* = 0.0553, Figure 1B). These data contrast with several previous reports [16,17,18] in which LAMP-2-ANCA titers, although not correlated with titers of MPO- or PR3-ANCA, were commonly detected in individuals also seropositive for either MPO- or PR3-ANCA.

Together, these results demonstrate that a greater majority of cases of pediatric-onset medium- and large-vessel vasculitis compared to small-vessel vasculitis in our cohort were not only seropositive for autoantibodies against LAMP-2 (exceeding 1 μg/mL) but had significantly elevated concentrations of circulating LAMP-2-ANCA; this was especially evident in cases of LVV. The observed differences in both LAMP-2-ANCA positivity (Table 2) and concentration in small-, medium- and large-vessel vasculitis was not associated with biological sex (Figure 1C, *p* = 0.8400) or the age of symptom onset (Figure 1D,E). Paradoxically, MVV and LVV are considered “ANCA-negative” subtypes of chronic PSV as they are rarely associated with autoantibodies against PR3 or MPO that are common to SVV.

### 2.2. LAMP-2-ANCA Seropositivity Is Associated with Lower Overall Disease Activity at Diagnosis

To gain insight into the potential for clinical utility of LAMP-2-ANCA measures in MVV and LVV, we asked if seropositivity or concentration of LAMP-2-ANCA at diagnosis could inform the present state of vasculitis-specific (inflammatory) activity or the ability to attain inactive disease within the first year following diagnosis. Results indicate that LAMP-2-ANCA concentration was not associated with generalized inflammation, as indicated by the concentration of C-reactive protein (CRP) or with disease-specific (inflammatory) activity at diagnosis measured by pVAS (Figure 2A). When considering seropositivity independently of LAMP-2-ANCA concentration, however, we observed significantly lower overall disease activity at diagnosis in LAMP-2-ANCA-seropositive (median pVAS = 15) versus -seronegative (median pVAS = 19) individuals (*p* = 0.0176), irrespective of the size of affected blood vessels (Figure 2B and Table 3).

Using a subset of participants with follow-up data after induction therapy (3–6 months post-diagnosis, n = 74) and one-year post-diagnosis (n = 70), we comparatively analyzed LAMP-2-ANCA seropositivity and concentration as being informative for disease trajectory based on two measures: overall improvement in disease activity (i.e., reduction in pVAS) and achievement of inactive disease. Chronic PSV patients showed marked improvement over the first 12 months of disease, with even the least improved patients achieving a minimum 33.3% decline in disease activity (pVAS). Individuals seronegative for LAMP-2-ANCA showed a greater improvement compared to LAMP-2-ANCA-seropositive individuals after induction therapy (Figure 2C left panel, *p* = 0.023) and one-year post-diagnosis (Figure 2C middle panel, *p* = 0.038). However, when the reduction in disease activity was calculated relative to pVAS at diagnosis (lower in LAMP-2-ANCA-seropositive patients, Figure 2B), no significant differences in improvement related to LAMP-2-ANCA status were observed (right panels in Figure 2C,D). Focusing on the achievement of inactive disease (pVAS ≤ 1), our results showed a comparable number of seropositive (13/21, 61.9%) and seronegative participants (7/10, 70.0%) with inactive disease or sustained disease activity one-year following diagnosis (Figure 2E right panel and Table 3) and no significant difference in LAMP-2-ANCA concentration at diagnosis between individuals (*p* = 0.5472, MVV + LVV, Figure 2E, left panel) that went on to achieve inactive disease one -year post-diagnosis.

Although our data show a relationship between LAMP-2-ANCA positivity and lower disease activity at diagnosis, neither diagnostic LAMP-2-ANCA seropositivity nor concentration was informative for overall disease activity following therapy induction or at one-year after diagnosis.

### 2.3. LAMP-2-ANCA Seropositivity and Concentration Are Not Significantly Associated with the Extent or Type of Organ Involvement

Given the observed relationship between LAMP-2-ANCA seropositivity and overall lower disease activity (pVAS) at diagnosis, we asked if the involvement of multiple, single, or particular organ systems were driving this association. In our cohort at the time of diagnosis, we observed 71.1% of patients with kidney involvement (renal pVAS ≥ 2), 37.7% of patients with upper respiratory tract involvement (ear, nose, throat (ENT) pVAS ≥ 2), 27.8% of patients with pulmonary system involvement (chest pVAS ≥ 2), and 23.4% of patients with cardiovascular involvement (cardio pVAS ≥ 2). A multivariate logistic regression model, adjusted for age at diagnosis and biological sex at birth, was used to evaluate the association between organ involvement and vessel size. As expected, we observed a significant association between vessel size and organ involvement. Compared to individuals with LVV, MVV less frequently affected the cardiovascular (OR 0.024, 95% CI 0.001–0.163, *p* = 0.001) and renal (OR 0.169, 95% CI 0.035–0.707, *p* = 0.019) systems; SVV had less cardiovascular (OR 0.023, 95% CI 0.004–0.097, *p* < 0.0001) system involvement, but higher involvement of the upper respiratory tract (URT) (OR 12.212, 95% CI 2.936–84.855, *p* = 0.003) and pulmonary system (OR 6.011, 95% CI 1.456–41.134, *p* = 0.027) involvement. The results of a regression model, adjusted for vessel size, assessing the relationship between the presence or absence of LAMP-2-ANCA and the involvement of more than one of these organ systems (*p* = 0.173) or a single organ (*p* = 0.281) were inconclusive. Furthermore, the results did not provide clear evidence of an association between LAMP-2-ANCA seropositivity/seronegativity and type of organ system involvement (Table 4).

We next observed significantly lower LAMP-2-ANCA concentrations in individuals with (mean LAMP-2-ANCA 1154.4 ng/mL) versus without (mean LAMP-2-ANCA 2498.6 ng/mL; *p* = 0.0043) URT involvement (Figure 3A), but not the pulmonary system as a whole (Figure 3B). In contrast, and despite no association between ANCA titers and hypertension (Figure 3D) [19], we observed significantly elevated LAMP-2-ANCA concentrations in individuals with cardiovascular involvement (mean LAMP-2-ANCA 3475.6 ng/mL) versus those individuals without this manifestation (mean LAMP-2-ANCA 1538.9 ng/mL; *p* = 0.0003) (Figure 3C). When using a multivariate linear regression model adjusted for vessel size, biological sex, and age at diagnosis, however, the results neither supported an association between the level of organ involvement (multi or single) and LAMP-2-ANCA concentration nor significant differences in LAMP-2-ANCA concentration between individuals with and without cardiovascular (*p* = 0.392), renal (*p* = 0.541), pulmonary (*p* = 0.511) and URT (*p* = 0.901) involvement.

## 3. Discussion

Chronic primary systemic vasculitis (PSV) is a heterogeneous group of diseases characterized by inflammation and damage to blood vessels that vary in size and location. Pediatric PSV affecting small blood vessels is commonly associated with autoantibodies (ANCA) against proteinase 3 (PR3) or myeloperoxidase (MPO). Herein, we demonstrate an increased prevalence and concentration of ANCA that are specific for LAMP-2 in children with PR3/MPO-ANCA-negative vasculitis affecting medium (MVV) and large (LVV) blood vessels. Our findings may prompt reconsideration of the presence and potential of monitoring autoantibody seropositivity against LAMP-2, or other autoantigens, in subtypes of MVV and LVV previously regarded as seronegative for ANCA.

PR3 and MPO are the predominant antigens of interest in small-vessel vasculitis and autoantibodies (ANCA) to these aid with classification of AAV subtypes granulomatosis with polyangiitis (GPA) and microscopic polyangiitis (MPA). There is conflicting data on the utility of ANCA titers to inform disease activity, and an as yet unknown role for MPO- or PR3-ANCA in organ-specific disease processes. Only recently, more than 40 years after their presence in vasculitis was discovered [20], a role for ANCA specificity towards PR3 and MPO in predicting disease outcomes in ANCA-associated SVV is being recognized. The presence of PR3-ANCA is associated with a higher risk of severe inflammatory lung disease, multi-organ involvement, and disease relapse; whereas MPO-ANCA are associated with more severe, renal-limited disease at presentation [21,22,23]. Not all patients, however, follow such patterns of disease, and there is evidence that other factors, such as type of organ involvement (e.g., renal versus non-renal) and genetic associations also impact risk [24,25]. ANCA specificity for LAMP-2 or other autoantigens, particularly those expressed in affected tissues, may have an additional prognostic role. Although the prevalence of LAMP-2-ANCA has been debated in PR3/MPO-ANCA-associated vasculitis (AAV) and AAV-related kidney disease, overlapping seropositivity for LAMP-2-ANCA with MPO- or PR3-ANCA in SVV is consistently observed [10,13,16,18,26].

The value of LAMP-2-ANCA seropositivity as a biomarker may be greatest among AAV patients who have MPO/PR3-ANCA-negative SVV, but more so in individuals with middle vessel vasculitis (MVV), namely, polyarteritis nodosa (PAN) and large vessel vasculitis (LVV), namely Takayasu’s arteritis (TAK). Due to the rarity of pediatric PAN and TAK combined with the absence of PR3- and MPO-ANCA, predictive biomarkers and specific disease activity markers are lacking. Determining levels of disease activity in PAN and TAK patients has proved challenging for several reasons: vascular imaging of medium- or large-sized vessels is frequently invasive and may not reliably differentiate between active inflammation versus damage; biopsy of affected vessels is often too risky and not feasible on a repeated basis; and traditional markers of inflammation (C-reactive protein and erythrocyte sedimentation rate) are non-specific and may not be elevated in organ-limited disease.

Our findings of elevated LAMP-2-ANCA seropositivity in pediatric PAN and TAK are consistent with reports in adults with PAN and TAK of elevated sera LAMP-2 protein [11,14], as well as elevated sera LAMP-2-ANCA [15,19]. In AAV, seropositivity for MPO or PR3-ANCA, rather than serial measures of titer, appear to have greater clinical value in predicting disease course [23,24,25,27]. In adult-onset PAN, Li et al. demonstrated a positive correlation between sera LAMP-2 protein concentration, C-reactive protein concentration, and overall disease activity in adult PAN [11]. The observed absence of a correlation between LAMP-2-ANCA titers and disease activity in this study was consistent with several previous reports on adult SVV [16,17,18] and our own observations in pediatric SVV [13]. Further, our work revealed that neither seropositivity nor titer at diagnosis was informative to disease course, with similar improvements in disease activity observed in LAMP-2-ANCA-seronegative and -seropositive participants.

In 2013, Kawakami et al. observed positive perinuclear (p) ANCA staining in adult cases of cutaneous polyarteritis nodosa and proposed that there are as yet undiscovered autoantigens [15]. More recently, Mukherjee et al. provided strong evidence for the presence of ANCA towards an unidentified antigenic target in sputa from adults with MPO/PR3-ANCA-negative eosinophilic granulomatosis with polyangiitis (EGPA). Importantly, their study demonstrated that ANCA reactivity was not observed in sera, emphasizing the importance of investigations in the affected tissue [28]. Additional studies support the existence of ANCA/autoantigens in MPO/PR3-ANCA-negative vasculitides including identification of elastase-ANCA in adult ANCA-negative glomerulonephritis [29] and the presence of alpha-enolase, a potential cytosolic autoantigen, in 82% of adults with predominantly ANCA-negative EGPA [30].

Our study has certain limitations. Despite having the largest reported pediatric chronic PSV study cohort, it may not be reflective of all populations. Further, the sample size may be underpowered for some analyses. Notably, it may not have been possible to observe any associations between organ system involvement and LAMP-2-ANCA, given that our multivariate logistic regression model did not reveal associations between MPO/PR3-ANCA status and particular organ systems as would be expected, given that a majority of cases have renal and pulmonary/URT involvement at diagnosis [31,32,33].

Distinguishing between the presence (or absence) and the antigenic target of autoimmune processes has important implications for therapeutic decision-making. Our data suggest that LAMP-2-ANCA may be important in childhood-onset PSV, where ANCA against classical targets, PR3 or MPO, are commonly absent. Although traditionally studied in the context of renal involvement, our data highlight a high prevalence and concentration of LAMP-2-ANCA in subtypes of PSV with extrarenal manifestations. Although substantive gaps in the understanding of LAMP-2-ANCA utility and pathogenicity remain, this report and others argue for continued examination of LAMP-2-ANCA, particularly in the context of medium- and large-vessel vasculitides that are commonly considered seronegative for autoantibodies.

## 4. Materials and Methods

### 4.1. Study Cohorts and Biosamples

The children and youth with chronic PSV that are described in this study (n = 90; Table 1) were enrolled in the Pediatric Vasculitis Initiative (PedVas), for which eligibility criteria have been described previously [32,34]. The Children’s and Women’s Research Ethics Board of the University of British Columbia [H12-00894] and the respective ethics review boards at participating PedVas centers gave their approval to the research protocol. Participants contributed blood in serum separation tubes (BD Biosciences, Franklin Lakes, NJ, USA) at the time of diagnosis. These sample were processed to sera according to a standard protocol and stored at −80 °C.

Individuals also included in the study were children and youth (n = 18; 38.9% female, median age of symptom onset 8.3 years, range 1.2–16.1 years) receiving care for a systemic autoinflammatory disease (SAID) characterized by recurrent episodes of uncontrolled inflammation in the absence of infection or autoimmunity (i.e., no detectable autoantibodies or autoreactive immune cells) [35]. Cohort characteristics are summarized in Appendix A. Approval from the Children’s and Women’s Research Ethics Board of the University of British Columbia was obtained to record diagnostic and demographic data in a research database (H14-00272) [36] and to collect and store (as described for PSV patients) sera (H15-00351). Sera were used to establish a range of LAMP-2-ANCA concentrations associated with systemic inflammation, as indicated by concentrations of human C-reactive protein (CRP ELISA kit) according to the manufacturer’s instructions (ThermoFisher, Waltham, MA, USA).

### 4.2. Clinical Data for Chronic PSV Participants

Clinical data were entered by participating centers on a web-based clinical data registry [31,32] and used to formally classify chronic PSV patients (n = 90) into disease subtypes under the broader designation of small- (SVV, n = 53), medium- (MVV, n = 16), and large- (LVV, n = 21) vessel vasculitis. A subset of relevant registry data is summarized in Table 1. The majority of cases of SVV met EULAR/PRINTO/PRES criteria for granulomatosis with polyangiitis (GPA, 43/53, 81.1%) [37,38]. MPA patients fulfilled the pediatric modified algorithm of the EMA [39] and EGPA patients met the ACR or Lanham criteria [37,40]. MVV subtypes included: polyarteritis nodosa (PAN, n = 11; EULAR/PRINTO/PRES criteria [37]), cutaneous PAN (cPAN, n = 3) and two cases of unclassified medium vessel vasculitis (uMVV, n = 2). Patients with LVV were predominantly Takayasu’s arteritis (TA, n = 19/21, 90.5%; EULAR/PRINTO/PRES criteria [37]) with two cases of unclassified large-vessel vasculitis (uLVV, n = 2). ANCA positivity and specificity for proteinase 3 (PR3) and myeloperoxidase (MPO) were entered in the registry by participating centers. As expected, the majority of cases of SVV were positive for MPO- or PR3-ANCA (49/52 cases with available data; 94.2%)

### 4.3. Quantification of Active Disease in Chronic PSV Participants

Disease activity at the time of sample/data collection (diagnosis) was calculated from the registry data using the pediatric vasculitis activity score (pVAS; range 0–63, where zero indicates inactive disease), which is a cumulative weighted score of disease activity across nine organ systems [41]. For individuals with follow-up data, improvement at one-year post-diagnosis was calculated as: % improvement = 100% × ((pVAS at diagnosis) − (pVAS at 1-year))/(pVAS at diagnosis). All participants had active disease at the time of diagnosis (and sample/data collection), with a median overall disease activity (median total pVAS) score equal to 17 (range 1–50). Using subcomponent scores of pVAS, analyses were focused on four critical organs/organ systems that drive treatment decisions early in the disease course and are frequently involved in SVV, MVV, and LVV, namely, kidneys (max. pVAS = 12), upper respiratory tract (max. pVAS = 6), pulmonary system (max. pVAS = 6), and cardiovascular system (max. pVAS = 6), where a subcomponent pVAS ≥ 2 indicated organ involvement [41].

### 4.4. Meso Scale Discovery (MSD) Electrochemiluminescence Assay to Detect Serum Anti-LAMP-2-Antibodies

Recombinant human (rh) LAMP-2 protein (R&D Systems, Minneapolis, MN, USA) was diluted in phosphate-buffered saline (PBS) and added (30 μL at 2.5 μg/mL) to 96-well electrochemiluminescence assay plates (Meso Scale Discovery, Rockville, MD, USA). Plates were sealed and incubated at 4 °C overnight, blocked (150 μL blocker A solution; RT, 1 h, 700 rpm) and washed (3 × 150 μL/well PBS + 0.05% Tween 20 (PBST)). A standard curve was generated with anti-human-LAMP-2 (H4B4) monoclonal antibody (BioLegend, San Diego, CA, USA, cat #354302) serially diluted from 1–4000 ng/mL in blocker A solution. Standards and sera (50 μL/well of 1:10 dilution) were added (1 h, RT, shaking), then wells were washed (3 × 150 μL/well PBST). A biotin-conjugated anti-IgG Fc (multi-species) antibody (ThermoFisher Scientific, Waltham, MA, USA, cat #7102852100) was added (50 μL/well at 1 μg/mL in blocker A; RT, 1 h, shaking) followed by strep-SULFO-TAG (50 μL/well at 0.5 μg/mL in blocker A; RT, 1 h, shaking) and 150 μL/well of MSD Gold Read buffer B with washing between steps (3 × 150 μL/well TBST). Plates were read using an MSD QuickPlex SQ 120 MM electrochemiluminescence instrument and a standard curve generated using a 4-parameter logistical curve fit algorithm within MSD Discovery Workbench Software (version 4.0.12). The assay was validated with sera, courtesy of Dr. Renate Kain at the University of Vienna, from young-onset ANCA-associated vasculitis patients known to be seropositive for anti-LAMP-2 antibodies [10,13,42] and a subset of participants with an SAID (n = 5) or a chronic PSV (n = 44; predominantly SVV) that were previously assessed for sera LAMP-2-ANCA concentrations using an indirect, in-house colorimetric ELISA [13]. Compared to the traditional colorimetric ELISA, the MSD platform displayed 10–60 times the linear range and improvement in the signal-to-noise ratio with a broad dynamic range. Using this assay, sera LAMP-2-ANCA concentrations in the majority (15/18; 83.3%) of individuals with an SAID were detected at less than or equal to 1000 ng/mL, with an average concentration of 661.0 ng/mL (range 44.4–1311.4 ng/mL) (Figure 1A) and informed the threshold of seropositivity for LAMP-2-ANCA at >1000 ng/mL in this study.

### 4.5. Statistical Analysis

GraphPad Prism v9.0 statistical software (GraphPad Software, La Jolla, CA, USA) was used to conduct the statistical analyses. Group differences were examined by one-way ANOVA and/or two-tailed *t*-tests. Pearson’s correlation coefficient was used to evaluate correlations. Where indicated, group differences were analyzed by Kruskal–Wallis, Mann–Whitney, or chi-squared statistical tests. R (R Foundation for Statistical Computing, version 4.2.2, Vienna, Austria) was used for multivariate analyses. A 95% confidence interval was used for all analyses, and a *p*-value equal to or less than 0.05 was regarded as significant.

## Figures and Tables

**Figure 1 ijms-25-03771-f001:**
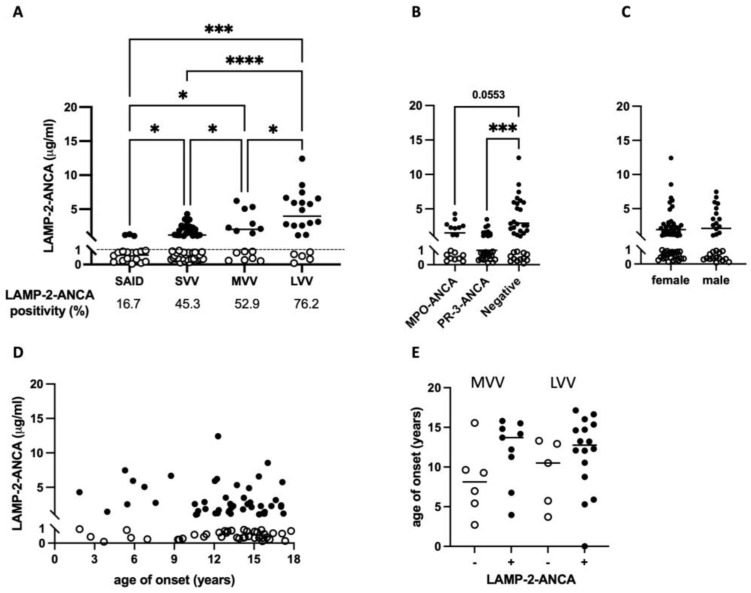
Sera LAMP-2-ANCA in chronic primary systemic vasculitis of childhood. (**A**) LAMP-2-ANCA concentration (*y*-axis; μg/mL) in sera from children and adolescents (*x*-axis) with systemic autoinflammatory disease (n = 18, SAID) and chronic primary systemic vasculitis (PSV) affecting small- (SVV, n = 53), medium- (MVV, n = 16) and large- (LVV, n = 21) sized blood vessels. Horizontal dotted line marks the threshold for LAMP-2-ANCA seropositivity in samples with greater than 1 μg/mL LAMP-2-ANCA. (**B**,**C**) LAMP-2-ANCA concentration (*y*-axis; μg/mL) in the pediatric chronic PSV cohort grouped (*x*-axis) based on (**B**) positivity for MPO-ANCA (n = 18), PR3-ANCA (n = 30), or neither MPO- or PR3-ANCA (negative, n = 35), and (**C**) biological sex. (**D**) Concentration of LAMP-2-ANCA (*y*-axis; μg/mL) in pediatric chronic PSV participants plotted against age of symptom onset (*x*-axis; years). (**E**) Age of symptom onset (*y*-axis; years) in LAMP-2-ANCA-seropositive and -seronegative patients with MVV and LVV (*x*-axis). Bars show means in (**A**–**C**), and medians in (**E**). LAMP-2-ANCA-seropositive (+) and -seronegative (−) samples indicated with closed and open circles, respectively. * Denotes statistical significance with *p* < 0.05, *** *p* < 0.001, **** *p* < 0.0001.

**Figure 2 ijms-25-03771-f002:**
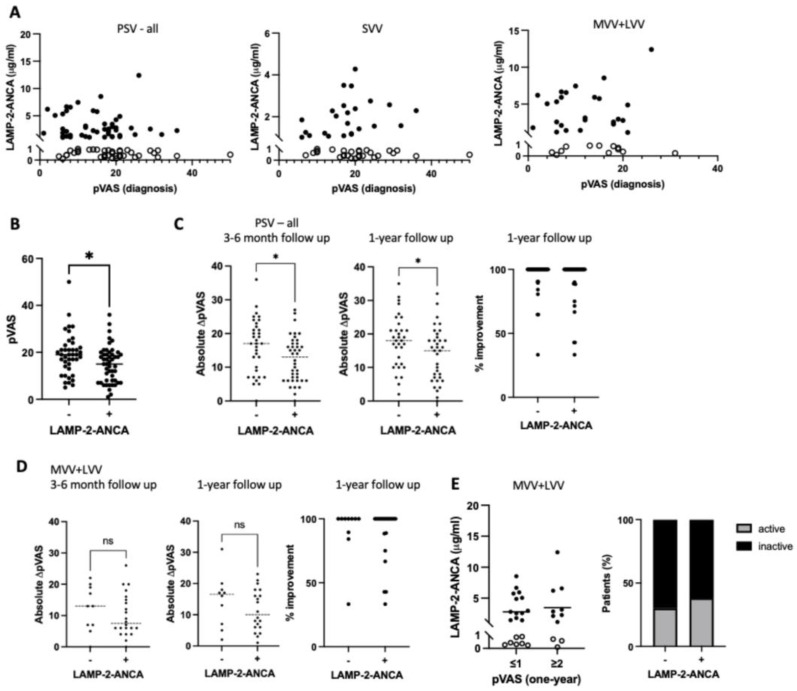
Measures of sera LAMP-2-ANCA positivity and concentration relative to disease activity at diagnosis and follow-up. (**A**) Concentration of LAMP-2-ANCA (*y*-axis; μg/mL) plotted against the pediatric vasculitis activity score (pVAS) (*x*-axis) at the time of diagnosis for all participants with chronic primary systemic vasculitis (PSV—all) and stratified according to the involvement of small (SVV) or medium- and large-sized (MVV + LVV) vessels. (**B**) Pediatric vasculitis activity score (pVAS) (*y*-axis, bar shows median) at diagnosis in LAMP-2-ANCA-seronegative (−) and -seropositive (+) participants (*x*-axis). (**C**) Reduction in disease activity between all LAMP-2-ANCA-seronegative (−) and -seropositive (+) PSV participants (*x*-axis) shown by absolute change in pediatric vasculitis activity score (pVAS) (*y*-axis) from diagnosis to (left panel) the completion of 3–6 months of induction therapy (seropositive, n = 40; seronegative, n = 34) and (middle panel) one-year follow-up (seropositive, n = 32; seronegative, n = 36). Improvement in disease activity expressed as percentage of maximal improvement from diagnosis to one-year follow-up. (**D**) Reduction in disease activity between LAMP-2-ANCA-seronegative (−) and seropositive (+) MVV/LVV participants (*x*-axis) shown by absolute change in pediatric vasculitis activity score (pVAS) (*y*-axis) from diagnosis to (left panel) the completion of 3–6 months of induction therapy (seropositive, n = 22; seronegative, n = 10) and (middle panel) one-year follow-up (seropositive, n = 20; seronegative, n = 10). Improvement in disease activity expressed as percentage of maximal improvement from diagnosis to one-year follow-up. (**E**) Concentration of LAMP-2-ANCA (*y*-axis; μg/mL, bar shows mean) at diagnosis in participants with medium-/large-sized vessel vasculitis (MVV + LVV) with inactive (pVAS at one-year ≤ 1) or active (pVAS at one-year ≥ 2) disease one-year later (left panel). Similarly (right panel), percentage (*x*-axis) of seronegative (−) and seropositive (+) MVV/LVV participants with inactive (black) or active (gray) disease one-year post-diagnosis. Bars show medians in (**B**,**C**) and means in (**D**). LAMP-2-ANCA-seropositive (+) and -seronegative (−) samples indicated with closed and open circles, respectively. ns indicates non-significant. * Denotes statistical significance with *p* < 0.05.

**Figure 3 ijms-25-03771-f003:**
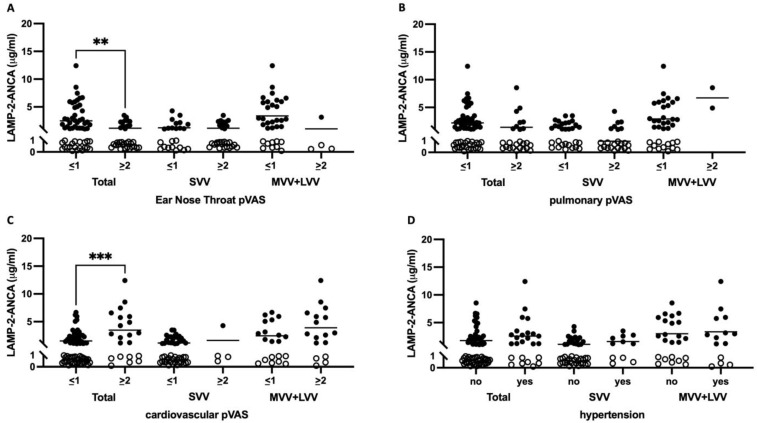
Organ system involvement and LAMP-2-ANCA titers in PSV. (**A**–**C**) Sera LAMP-2-ANCA concentration (*y*-axis; μg/mL) versus disease activity (component pVAS ≤ 1 is inactive and ≥2 is active) in (**A**) upper respiratory tract (ear–nose–throat component of pVAS), (**B**) pulmonary system, (**C**) cardiovascular system, or relative to (**D**) the presence of hypertension (yes/no) in children and adolescents with chronic primary systemic vasculitis (*x*-axis, total, n = 25) affecting small (SVV, n = 11) and medium/large (MVV + LVV, n = 14) vessels. LAMP-2-ANCA-seropositive and -seronegative samples indicated with closed and open circles, respectively. Bars show means. ** Denotes statistical significance with *p* < 0.01, *** *p* < 0.001.

**Table 1 ijms-25-03771-t001:** Pediatric chronic primary systemic vasculitis (PSV) cohort.

	Small Vessel (n = 53)	Medium Vessel (n = 16)	Large Vessel (n = 21)
*PSV subtype*, n (%)			
GPA/limGPA	43 (81.1)	-	-
MPA	5 (9.4)	-	-
EGPA	2 (3.8)	-	-
ANCA + GN	1 (1.9)	-	-
PAN/cPAN	-	14 (87.5)	-
TA	-	-	19 (90.5)
UPV	2 (3.8)	2 (12.5)	2 (9.5)
*Sex*, n (%)			
Female	35 (66.0)	10 (62.5)	15 (71.4)
Male	18 (34.0)	6 (37.5)	6 (28.6)
No significant difference in representation of males/females between groups (*p* = 0.8400) based on chi-squared test			
*Age (years) of symptom onset*—median (range)	* 14.2 (1.9–17.3)	11.2 (2.7–15.8)	12.6 (3.7–17.1)
^1^ *ANCA Antigen Positivity*, n (%)			
Proteinase 3 (PR3)	29 (55.8)	1 (6.7)	-
Myeloperoxidase (MPO)	18 (34.6)	-	-
PR3 and MPO	2 (3.8)	-	-
Neither	3 (5.8)	14 (93.3)	18 (100)
*Disease activity (pVAS) at diagnosis*			
Total pVAS, median (range)	19 (6–50)	8 (1–18)	16 (6–26)
Subcomponent pVAS, median (range)			
Renal	10 (0–12)	0 (0–6)	4 (0–12)
Cardiovascular	0 (0–4)	0	4 (0–6)
Pulmonary	0 (0–6)	0 (0–4)	0 (0–6)
Upper respiratory tract	4 (0–6)	0 (0–6)	0 (0–4)
^2^ *Induction Treatment*, n (%)			
^3^ Immune-suppressing agents	38 (73.0)	4 (25.0)	2 (9.5)
^4^ Disease-modifying agents	6 (11.5)	6 (37.5)	11 (52.4)
^5^ Biologic agents	-	1 (6.3)	6 (28.6)
Corticosteroids	48 (92.3)	12 (75.0)	20 (95.2)
^2^ *Maintenance Treatment*, n (%)			
^3^ Immune-suppressing agents	16 (37.2)	4 (28.6)	4 (20.0)
^4^ Disease-modifying agents	14 (32.5)	9 (64.3)	10 (50.0)
^5^ Biologic agents	-	2 (14.3)	8 (40.0)
Corticosteroids	41 (95.3)	10 (71.4)	19 (95.0)

^1^ Percentage of patients with ANCA calculated based on available data for 52/53 SVV cases, 15/16 MVV cases and 18/21 LVV cases, ^2^ Percentage of patients on each induction treatment (for 3–6 months following diagnosis) is calculated based on available data within the SVV (52/53), MVV (16/16), and LVV (21/21) groups and likewise for maintenance treatment (initiated after induction treatment is complete) within the SVV (43/53), MVV (14/16), and LVV (20/21) groups. ^3^ Immune-suppressing agents are most commonly cyclophosphamide and rituximab. ^4^ Disease-modifying agents are most commonly azathioprine, methotrexate, and mycophenolate mofetil. ^5^ Biologic agents are most commonly infliximab and tocilizumab. GPA, granulomatosis with polyangiitis; limGPA, limited GPA; MPA, microscopic polyangiitis; EGPA, eosinophilic GPA; GN, glomerulonephritis; PAN, polyarteritis nodosa; cPAN, cutaneous PAN; TA, Takayasu’s arteritis; UPV, unclassifiable primary vasculitis; pVAS, pediatric vasculitis activity score. * Significant difference in median age between SVV and MVV (*p* = 0.0209), but not SVV and LVV (*p* = 0.0666) or MVV and LVV (*p* = 0.5357) based on Mann–Whitney tests.

**Table 2 ijms-25-03771-t002:** LAMP-2-ANCA seropositivity in pediatric chronic primary systemic vasculitis (PSV).

	PSV Total (n = 90)		Small Vessel (n = 53)		Medium Vessel (n = 16)		Large Vessel (n = 21)	
^a^ *LAMP-2-ANCA*	−	+	−	+	−	+	−	+
n	n = 41	n = 49	n = 29	n = 24	n = 7	n = 9	n = 5	n = 16
%	45.6%	54.4%	54.7%	45.3%	41.2%	52.9%	23.8%	* 76.2%
^b^ *Sex*, n (%)								
Female	26 (63.4)	34 (69.4)	18 (62.1)	17 (70.8)	4 (57.1)	6 (66.7)	4 (80.0)	11 (68.8)
Male	15 (36.6)	15 (30.6)	11 (37.9)	7 (29.2)	3 (42.9)	3 (33.3)	1 (20.0)	5 (31.2)
^c^ *Age of onset*, years								
Median	13.8	13.2	14.5	13.1	8.1	13.7	10.5	13.2
Range	1.9–17.3	1.9–17.2	1.9–17.8	1.9–17.2	2.7–15.5	4.0–15.8	3.7–13.3	5.3–17.1
^d^ *ANCA antigen positivity*, n (%)								
PR3	19 (22.4)	11 (12.9)	18 (34.6)	11 (21.2)	1 (5.9)	−	−	−
MPO	9 (10.6)	9 (10.6)	9 (17.3)	9 (17.3)	−	−	−	−
PR3 and MPO	1 (1.2)	1 (1.2)	1 (1.9)	1 (1.9)	−	−	−	−
Neither	12 (14.1)	23 (27.1)	1 (1.9)	2 (3.8)	6 (35.3)	8 (47.1)	5 (27.8)	13 (72.2)

^a^ LAMP-2-ANCA-seronegative indicated by (−) and -seropositive indicated by (+). ^b^ Percentage calculated from the total seronegative or seropositive participants of the same vessel size; no statistically significant differences (chi-squared test) between seropositive and seronegative individuals in the PSV (0.5494), SVV (0.5024), MVV (0.6963), and LVV (0.6269) groups. ^c^ No statistically significant differences between seropositive and seronegative individuals in the PSV (0.7235, Mann–Whitney test), SVV (0.1479, Mann–Whitney test), MVV (0.1146, unpaired *t*-test), and LVV (0.1085, unpaired *t*-test) groups. ^d^ Percentage calculated based on available data for participants within the PSV (85/90), SVV (52/53), MVV (15/16), and LVV (18/21) groups. * Significant difference in the prevalence of seropositive individuals between SVV and LVV (*p* = 0.0162) based on chi-squared test. Across all groups, *p* = 0.0545 for the prevalence of seropositivity; SVV vs. MVV, *p* = 0.4415; MVV vs. LVV, *p* = 0.1993.

**Table 3 ijms-25-03771-t003:** Disease activity at diagnosis and 1-year follow up stratified by LAMP-2-ANCA seropositivity.

	PSV Total		Small Vessel		Medium Vessel		Large Vessel	
^a^ *LAMP-2-ANCA*	−	+	−	+	−	+	−	+
*Disease activity (total pVAS)*, median (range)								
^b^ at diagnosis	19 (5–50)	* 15 (1–36)	20 (7–50)	18 (6–36)	8 (5–31)	7 (1–18)	19 (6–20)	15 (6–26)
^c^ at 1-year	0 (0–4)	0 (0–12)	0 (0–6)	0 (0–12)	0 (0)	1 (0–4)	2 (3–4)	0 (0–12)
^d^ Inactive disease (total pVAS ≤ 1), n (%)								
at 1-year	26 (37.1)	25 (35.7)	19 (48.7)	12 (30.8)	5 (38.5)	4 (30.8)	2 (11.1)	9 (50.0)

^a^ LAMP-2-ANCA-seronegative indicated by (−) and seropositive indicated by (+). ^b^ At diagnosis, pVAS available for n = 90 PSV patients. * Significant difference in disease activity (*p* = 0.0176) across all groups based on unpaired *t*-test. ^c^ At 1 year post-diagnosis, pVAS available for n = 70 PSV patients: n = 39 SVV (15 LAMP2 ANCA+), n = 13 MVV (8 LAMP2 ANCA+), n = 18 LVV (13 LAMP2 ANCA+). ^d^ Inactive disease defined as total pVAS ≤ 1 at one-year post-diagnosis. Percentage is relative to the total number of seronegative or seropositive participants of the same vessel size.

**Table 4 ijms-25-03771-t004:** Association between organ system involvement and LAMP-2-ANCA seropositivity.

Organ System	LAMP-2-ANCA		^b^ Adjusted OR (95% CI)	Standard Error	*p*-Value
	Seronegative (n = 36) ^a^	Seropositive (n = 41) ^a^			
Cardiovascular	7 (19.4%)	14 (34.1%)	0.82 (0.15, 3.73)	0.79	0.800
Renal	29 (81.0%)	35 (85.4%)	1.70 (0.46, 6.91)	0.68	0.429
Pulmonary	16 (44.4%)	9 (22.0%)	0.49 (0.16, 1.46)	0.56	0.198
URT	19 (52.8%)	13 (31.7%)	0.58 (0.19, 1.79)	0.57	0.339

^a^ Number of PSV patients with involvement of URT, cardiovascular, renal, and pulmonary systems. ^b^ The reference category is LAMP-2-ANCA seronegativity. OR; odds ratio, CI; confidence interval, URT; upper respiratory tract.

## Data Availability

The data that support the findings of this study are available from the corresponding author upon reasonable request.

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
