# Peer review of "Anti-LAMP-2 Antibody Seropositivity in Children with Primary Systemic Vasculitis Affecting Medium- and Large-Sized Vessels"

_ijms, 2024, doi:10.3390/ijms25073771_

Round 1
Reviewer 1 Report
Comments and Suggestions for Authors
The authors report a high prevalence and concentrations of LAMP-2-ANCA in pediatric patients with cronic primary systemic vasculitis, seronegative for traditional ANCA testing and affecting large vessels. The main question is the involvement of Anti-LAMP2 antibodies in a subset of pediatric patients with primary systemic vasculitis (PSM). This is original work, relevant to the field. The pathophysiology of PSM is still not elucidated. The results show higher prevalence and concentration of LAMP-2-ANCA in chronic PSV, affecting large-sized blood vessels.
The study points to the roles of autoantigens other than proteinase 3 and myeloperoxidase in children with primary systemic vasculitis, particularly in medium and large-sized blood vessels.Abstract shoud be re-written, it is not undertsndable in this format due to language errors.
Materials and Methods section should be placed before Results.
Discussion section should be enlarged due to a large number of results obtained in the study.
The conclusions are consistent; the methodology is appropriate and the number of patients/sera (90) sufficient.
The reference list is appropriate.
Author Response
Reviewer 1 Comment:
The authors report a high prevalence and concentrations of LAMP-2-ANCA in pediatric patients with cronic primary systemic vasculitis, seronegative for traditional ANCA testing and affecting large vessels. The main question is the involvement of Anti-LAMP2 antibodies in a subset of pediatric patients with primary systemic vasculitis (PSM). This is original work, relevant to the field. The pathophysiology of PSM is still not elucidated. The results show higher prevalence and concentration of LAMP-2-ANCA in chronic PSV, affecting large-sized blood vessels.
The study points to the roles of autoantigens other than proteinase 3 and myeloperoxidase in children with primary systemic vasculitis, particularly in medium and large-sized blood vessels.
Thank you very much for taking the time to review this manuscript. Please find the detailed responses below and the corresponding revisions/corrections highlighted/in track changes in the re-submitted files.
Reviewer 1 Comment:
Abstract shoud be re-written, it is not undertsndable in this format due to language errors.
Response:
Thank you for pointing out this oversight. We have modified our abstract as below:
‘Chronic primary systemic vasculitis (PSV) comprises a group of heterogeneous diseases that are broadly classified by vessel size, clinical traits and the presence (or absence) of anti-neutrophil cytoplasmic antibodies (ANCA) against proteinase-3 (PR3) and myeloperoxidase (MPO). In small vessel vasculitis (SVV), ANCA are not present in all patients, and they are rarely detected in patients with vasculitis involving medium (MVV) and large (LVV) sized vessels. Some studies have demonstrated that lysosome associated membrane protein-2 (LAMP-2/CD107b) is a target of ANCA in SVV, but its presence and prognostic value in childhood MVV and LVV is not known. This study utilized retrospective sera and clinical data obtained from 90 children and adolescents with chronic PSV affecting small- (SVV, n = 53), medium- (MVV, n = 16) and large- (LVV, n = 21) sized blood vessels. LAMP-2-ANCA was measured in time-of-diagnosis sera using a custom electrochemiluminescence assay. The threshold for seropositivity was established in a comparator cohort of patients with systemic autoinflammatory disease. The proportion of LAMP-2-ANCA seropositive individuals and sera concentrations of LAMP-2-ANCA were as-sessed for associations with overall and organ-specific disease activity at diagnosis and one-year follow up.This study demonstrated a greater time-of-diagnosis prevalence and sera concentration of LAMP-2-ANCA in MVV (52.9% seropositive) and LVV (76.2%) compared to SVV (45.3%). Further, LAMP-2-ANCA seropositive individuals had significantly lower overall, but not organ-specific, disease activity at diagnosis. This did not, however, result in a greater reduction in disease activity or the likelihood of achieving inactive disease one year after diagnosis. The results of this study demonstrate particularly high prevalence and concentration of LAMP-2-ANCA in chronic PSV that affects large sized blood vessels and is seronegative for traditional ANCA. Our findings invite reconsideration of roles for autoantigens other than MPO and PR3 in pediatric vasculitis, particularly in medium and large sized blood vessels.’
Reviewer 1 Comment:
Materials and Methods section should be placed before Results.
Response:
We thank the reviewer for this suggestion. According to the guidelines of the International Journal of Molecular Science, they state it should be placed after the discussion section, so we could not modify it.
Reviewer 1 Comment:
Discussion section should be enlarged due to a large number of results obtained in the study.
Response:
Thank you for this comment. We have added the following paragraphs to the discussion section.
PR3 and MPO have been, and continue to be, the predominant antigens of interest in small vessel vasculitis despite known limitations in cleanly defining AAV subtypes granulomatosis with polyangiitis (GPA) and microscopic polyangiitis (MPA), conflicted data on the utility of ANCA titers to inform disease activity, and an as yet unknown role for MPO- or PR3-ANCA in organ-specific disease processes. Only recently and more than 40 years after their presence in vasculitis was discovered [20], a role for ANCA specificity towards PR3 and MPO in predicting disease outcomes in ANCA-associated SVV is becoming increasingly accepted. The presence of PR3-ANCA is associated with a higher risk of severe inflammatory lung disease, multi-organ involvement, and disease relapse; whereas MPO-ANCA is associated with renal limited disease [21-23]. Not all patients, however, follow such patterns of disease and there is evidence that other factors, such as type of organ involvement (e.g. renal versus non-renal) and genetic associations also impact risk [24,25]. ANCA specificity for other autoantigens, particularly those expressed in affected tissues, may further contribute to the refinement of this model. For example, in patients with ANCA against traditional target antigens PR3 or MPO, does the presence or absence of autoantibodies against LAMP-2 improve prediction of disease outcome? Although the prevalence of LAMP-2-ANCA has been debated in PR3-/MPO-ANCA-associated vasculitis (AAV) and AAV-related kidney disease, overlapping seropositivity for LAMP-2-ANCA with MPO- or PR3-ANCA in SVV is consist-ently observed [10,13,16,18,26].
The value of LAMP-2-ANCA seropositivity, however, may be greatest in the groups of vasculitis patients that are seronegative for MPO-/PR3-ANCA, inclusive of ANCA-negative SVV, but more so in individuals with larger vessel diseases namely, polyarteritis nodosa (PAN) and Takayasu’s arteritis (TA). Due to the rarity of pediatric PAN and TA combined with the absence of PR3- and MPO-ANCA, predictive biomarkers and specific disease activity markers are lacking. Determining levels of disease activity in PAN and TA patients has proved challenging for several reasons: vascular imaging of medium or large vessels is frequently invasive and may not reliably differentiate between active inflammation versus damage; biopsy of affected vessels is often too risky and not feasible on a repeated basis; traditional markers of inflammation (C-reactive protein and erythrocyte sedimentation rate) are non-specific and may not be elevated in organ-limited disease.
Reviewer 1 Comment:
The conclusions are consistent; the methodology is appropriate and the number of patients/sera (90) sufficient. The reference list is appropriate.
Response:
Thank you for your comments.
Reviewer 2 Report
Comments and Suggestions for Authors
Reviewing an exciting article about the possible involvement of Anti-LAMP2 antibodies in a subset of patients with primary systemic vasculitis has been a great pleasure. New insights are essential to advancing understanding, especially in heterogenic diseases. The text is intriguing and captivates the reader to the end.
I have only minor remarks:
1. Lines 33-40: Please check the spaces between words and abbreviations
2. Lines 102-103: Please refer to the appropriate Table number. At the same time, the reader can follow the tables more naturally, numbers from 1 to 4, through the text, similarly to how you have done well in the presentation of figures.
3. Table 1: Age. Please use one type of presentation for Range. Proposal: As you did at Total pVAS - (Range). It is more consistent and saves space.
Author Response
Reviewer 2:
Reviewing an exciting article about the possible involvement of Anti-LAMP2 antibodies in a subset of patients with primary systemic vasculitis has been a great pleasure. New insights are essential to advancing understanding, especially in heterogenic diseases. The text is intriguing and captivates the reader to the end.
Thank you very much for taking the time to review this manuscript. Please find the detailed responses below and the corresponding revisions/corrections highlighted/in track changes in the re-submitted files.
Reviewer 2 Comment:
Lines 33-40: Please check the spaces between words and abbreviations
Response:
We thank the reviewer for this oversight.. All documents have been checked and corrected.
Reviewer 2 Comment:
Lines 102-103: Please refer to the appropriate Table number. At the same time, the reader can follow the tables more naturally, numbers from 1 to 4, through the text, similarly to how you have done well in the presentation of figures.
Response:
We thank the reviewer for this suggestion. We have thoroughly reviewed the manuscript and referred to the appropriate table numbers where necessary.
Reviewer 2 Comment:
Table 1: Age. Please use one type of presentation for Range. Proposal: As you did at Total pVAS - (Range). It is more consistent and saves space.
Response:
We thank the reviewer for this suggestion. We agree this would help make the graph more consistent and compact. These changes have been made to Table 1.